# Machine Vision Algorithm for Identifying Packaging Components of HN-3 Arterial Blood Sample Collector

**Zhendong Shang [1,2,*], Qinzhang Wei [1,2] and Zhaoying Li [1,2]**

1    School of Mechanical and Electrical Engineering, Henan University of Science and Technology, Luoyang 471003, China; qinzhangwei65@gmail.com (Q.W.); lizhaoying09@gmail.com (Z.L.)
2    Henan Province Key Laboratory of Mechanical Design and Transmission System, Luoyang 471003, China
\*    Correspondence: jdszd@haust.edu.cn; Tel.: +86-135-2698-8202

**Abstract:** The arterial blood sample collector produced in large quantities often fails to meet the requirements due to missing components in the packaging bag, and traditional manual detection methods are both inefficient and inaccurate. To solve this problem, a PyCharm-integrated development environment was used to study image processing and recognition algorithms for identifying components inside the packaging bag of the HN-3 arterial blood sample collector. The machine vision system was used to capture images of the packaging bags of the HN-3 Arterial blood sample collector. Template matching was employed to extract the packaging ROI, and the threshold segmentation method in the HSV color model was used to extract material features based on the packaging ROI. Morphological processing algorithms such as dilation or erosion were used to enhance the connectivity of the extracted features. The existence of components was determined by setting thresholds for the connected domain area or length. The results of the recognition experiment show that the false detection rate is 0.2%, the missed detection rate is 0%, and the average image processing time per product is no more than 39 ms. Compared with manual recognition methods, the efficiency and accuracy have been improved by 36.5 times and 2.3%, respectively. The experimental results confirm the effectiveness of the image processing algorithm.

**Keywords:** arterial blood collection device; machine vision; ROI extraction; HSV color model





## 1. Introduction

Arterial blood gas analysis is a vital method used to measure various indicators in human arterial blood, including carbon dioxide pressure, pH, and oxygen pressure. It plays a crucial role in diagnosing and treating respiratory failure and acid-base imbalances in patients [1]. The HN-3 disposable human arterial blood sample collector, produced by Henan Tuoren Medical Device Co., Changyuan China, is a widely used medical tool for collecting arterial blood samples and has a high market share. Figure 1 illustrates the components of the HN-3 collector, which include an arterial blood collection needle (consisting of a sample storage device and a blood collection needle head), a needle plug, and a cone head seal. To facilitate clinical application, all components are packed in a plastic bag. Since arterial blood sample collectors are classified as Class III medical devices, it is essential to perform thorough inspections to ensure that no components are missing from each product packaging bag. However, traditional manual inspection methods are prone to missed or false detections due to factors such as visual fatigue among workers, resulting in low detection efficiency. Therefore, there is an urgent need for a machine-vision-based system to detect missing materials in arterial blood sample collectors. The key challenge lies in accurately identifying the packaging components of the arterial blood sample collector.

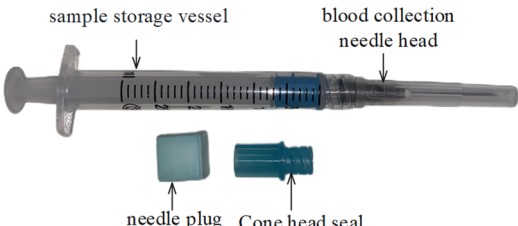

**Figure 1.** Composition of arterial blood collector.

Performing automated machine vision inspection using image processing techniques is a common practice in industrial settings to ensure product quality before packaging and shipment to customers [2]. Several studies have focused on quality inspection algorithms and defect recognition systems based on machine vision technology for various products, such as ampoule packaging [3], online packaging defect recognition [4], and detecting adulterated candy in bag packaging using Raman hyperspectral imaging [5]. However, there have been no reports on component missing inspection specifically for arterial blood sample collectors after packaging.

This article uses PyCharm as the development environment to study the arterial blood sample collector packaging component missing detection algorithm based on machine vision. The PyCharm-integrated development environment is configured with software packages such as Opencv4.4.0 and modbus_tk that depend on the Python language. Opencv software package is used for image processing operations, and Modbus-TCP protocol is implemented using the Modbus_tk software package to achieve data exchange between the PC and the sorting system PLC. The image algorithm is divided into three modules: packaging image ROI (region of interest) extraction, threshold segmentation, and component recognition. The packaging ROI extraction module locates key areas in the image to ensure that image processing focuses on the component area, thus increasing feature segmentation reliability. The threshold segmentation module converts the extracted ROI image from the RGB color space to the HSV color space and divides it into H, S, and V channels. These channels are then used to extract the characteristics of the plunger, drawbar piston, cone head seal, and blood collection needle tip based on thresholding techniques. The comprehensive judgment module determines the inspection result based on the area value and distance value of the connected components. By developing this machine-vision-based algorithm, the aim is to address the issue of components missing from inspection in arterial blood sample collectors after packaging, providing a reliable solution for quality control in the manufacturing process.

This algorithm utilizes advanced image processing techniques to effectively analyze and process images, resulting in the precise identification of components. The evaluation of experimental results has showcased the algorithm's exceptional accuracy and rapid processing time, thereby establishing its feasibility and practicality for real-world applications. The outcomes of this research hold considerable significance for the arterial blood gas analysis market in China, offering medical device manufacturers and healthcare institutions a valuable opportunity to enhance product quality and work efficiency. By improving the accuracy and efficiency of packaging component recognition, this algorithm has laid the foundation for substantial progress in the industry.

## 2. ROI Extraction

Accurately and efficiently extracting ROIs from images can reduce computational complexity [6]. In machine vision image processing, there are various methods for ROI extraction, such as machine learning and template matching. Machine learning can provide powerful object detection capabilities [7], but often requires a large amount of annotated data for sample training, making it difficult to meet the conditions in many cases. In contrast, template matching methods can bypass manual annotation and determine the most likely location of a single template image in a target image with minimal computational cost [8].

When identifying the presence or absence of arterial blood sample collector packaging components in an image, it is necessary to locate the ROI of the packaging image since the range of component locations can be controlled within the packaging background area.

In template matching, pixel-level matching is used to determine the position of a template image within a target image. By sliding the template image over different positions of the target image and calculating the similarity between them, the best-matching pixel position can be found.

Specifically, the template and target images are first converted to grayscale for pixel-level matching. The template image is then slid over every possible position in the target image, and the similarity between pixels at each sliding position is measured using methods like normalized squared difference. This involves calculating the squared differences and normalizing them to obtain similarity values. These similarity values are organized into a matching result matrix, which has the same size as the target image. Each element of the matrix represents the similarity measure at the corresponding position. The best matching position in the result matrix is typically determined by finding the minimum or maximum value, which corresponds to the vertex or center position of the template image in the target image. Pixel-level matching allows for accurate localization of the template image within the target image, enabling ROI extraction.

The specific operation steps are as follows:

First, the template image and the image to be matched need to be passed into the cv2.matchTemplate function to make the template image in the image match the sliding calculation; this is developed through the cv2.TM_SQDIFF_NORMED normalized squared difference matching method, which calculates the squared difference value of each sliding. Each calculation results will be organized into the form of a matrix.

Secondly, the minimum value in the result matrix is counted using the cv2.minMaxLoc function, and the minimum coordinate point is the coordinate of the upper left corner of the ROI. The shape function is used to calculate the width and height (w, h) of the template image and to add them with the minimum coordinate to obtain the coordinate point of the lower right corner of the ROI.

Finally, the two points of the horizontal and vertical coordinate values are calculated through two-dimensional array slicing (to extract the data in the matrix with the coordinates as the index number) to obtain the packing image ROI; the cv2.Rectangle function is used to draw the ROI region in the original image.

In this paper, we use the principle of normalized correlation template matching to determine the degree of image matching and find the matching position based on the grayscale information of the image, using the correlation function between the matching sub-image and the grayscale value of the template image. The results of image processing template matching are shown in Figure 2.

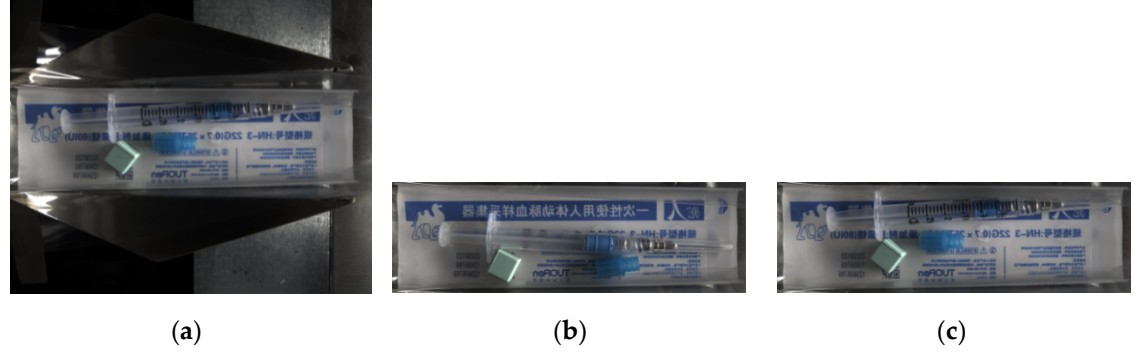

(**a**)                                        (**b**)                                        (**c**)

**Figure 2.** Template matching result. (**a**) Source image; (**b**) template image; (**c**) ROI image.

## 3. Threshold Segmentation

During the extraction of features for color image segmentation, the non-uniformity of the RGB color model can often lead to confusion and loss of color information [9]. In contrast, the HSV color model has stronger uniformity and more distinct components than the RGB color model, making it better suited to reflect specific color information. It visually expresses the hue, brightness, and saturation between colors, making it more suitable for color image segmentation [10].

Image noise refers to unwanted interference or random disturbances present in digital images. It can be introduced during the processes of image acquisition, transmission, storage, and processing, resulting in a degradation of image quality and loss of information. Therefore, prior to performing feature segmentation on the ROI image, it is usually necessary to apply image-filtering techniques to reduce the interference caused by noise points and improve the accuracy of subsequent image segmentation. The filtering method used is the cv2.gaussianblur function in Python-OpenCV.

*H*, *S*, and *V*, respectively, represent hue, saturation, and value. As OpenCV reads the initial image in the BGR color space, it is necessary to convert it to the HSV color space.

The most commonly used method for image segmentation based on the HSV color model is threshold segmentation, which usually involves using a global threshold to divide the image into a binary one. For color images with complex content, the concept of threshold segmentation can be extended to 3D space. Since the three components of the HSV color space are independent from each other, the problem of segmenting a three-dimensional space can be transformed into three independent one-dimensional segmentation problems. The *H*, *S*, and *V* components of the packaging image for the arterial blood sample collector are shown in Figure 3.

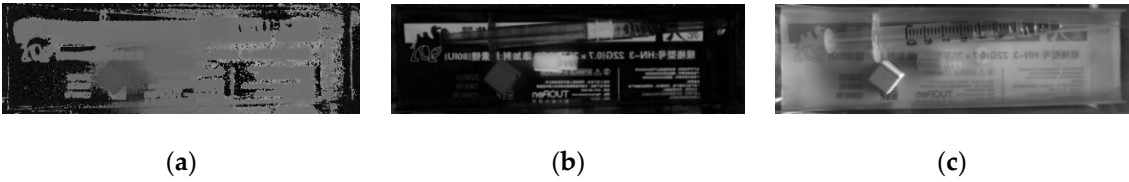

(**a**)          (**b**)          (**c**)

**Figure 3.** HSV three components. (**a**) *H* component; (**b**) *S* component; (**c**) *V* component.

By utilizing upper and lower threshold values for segmenting features in each of the three components of HSV separately, the cv2.createTrackbar function module can be employed. This module serves as a tool for dynamically adjusting parameters associated with an image window. Through continuous adjustment of the upper and lower limits of the H, S, and V components, while observing the dynamic segmentation results of the cv2.inRange function module, the upper and lower limits are determined based on the saliency of the segmentation features. The intersection of the segmentation results from each channel is then taken to obtain the feature region, as shown in Equation (1).

$$\text{Region} = \text{region\_S} \cap \text{region\_V} \cap \text{region\_H} \tag{1}$$

## 4. Component Recognition

### 4.1. Needle Plug Identification

Based on the template matching of the image, the features of the needle plug are segmented and evaluated. Through multiple experiments, it was determined that setting the upper threshold as np.array(90,70,255) and the lower threshold as np.array(60,31,45) achieved a notable segmentation effect, effectively distinguishing the needle plug feature from other background regions in the image. The resulting image after HSV threshold segmentation is depicted in Figure 4.

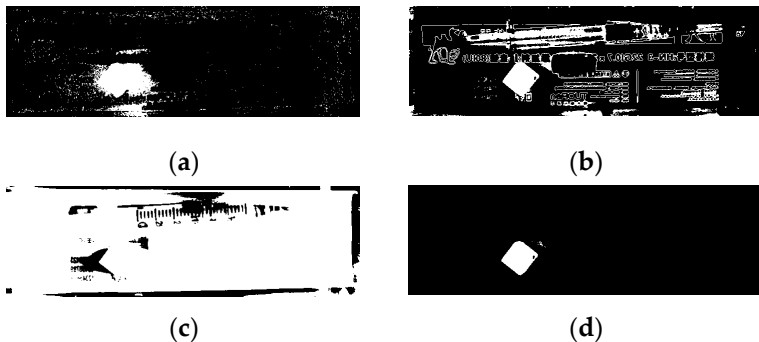

**Figure 4.** Threshold segmentation image. (**a**) The result of channel separation for *H*; (**b**) the result of channel separation for *S*; (**c**) the result of channel separation for *V*; (**d**) the result of comprehensive segmentation.

The features of segmentation can be slightly disturbed by factors such as component location and package reflection, resulting in inaccurate contour extraction. Performing a morphological opening operation on the integrated segmentation result image, which means first eroding and then dilating the image, can eliminate the slight disturbances around the main features without significantly changing the area of the connected domain of the main feature. The image after morphology processing is shown in Figure 5.

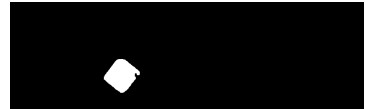

**Figure 5.** Morphologically processed image.

According to the result of the opening operation, the cv2.findContours function is used to find the contours of connected regions, the area of the connected region is calculated using cv2.contourArea, denoted as Area(r), and then it is compared with the set threshold value $T_a$ for needle plug. If the area is less than the threshold, then there is no needle plug; otherwise, a needle plug exists, as shown in Equation (2).

$$\text{Area}(r) \geq \text{Area}_{Ta} \tag{2}$$

### 4.2. Identification of a Sample Storage Container and Cone Head Seal

As the sample storage container comprises a drawbar piston and an outer shell, the detection of a missing sample storage container can be transformed into the detection of a missing drawbar piston, considering its distinct color. However, during the segmentation of the cone head seal and drawbar piston features, adhesion may occur due to their similar colors when they are in close proximity. To address this issue, a method based on a connected domain area threshold classification is employed for detection and judgment.

After examining the filtered HSV channel images, it was observed that the target feature grayscale in the H and V components exhibited low contrast with the surrounding grayscale, while the contrast was higher in the S component. Hence, it is preferable to initially segment the feature from the S component and subsequently eliminate interference within the S component using other components. Through multiple trials, it was determined that the most effective feature segmentation occurred when the upper threshold for the S component was set to 198 and the lower threshold was set to 125. The segmentation results are illustrated in Figure 6. Further segmentation with the H and V components, along with the intersection with the S component, allows for the removal of irrelevant white regions in the S component. This is accomplished by adjusting the upper and lower thresholds of the H component. Specifically, setting the upper threshold of the H component to 109 and the lower threshold to 96 achieves the desired segmentation, as depicted in Figure 7.

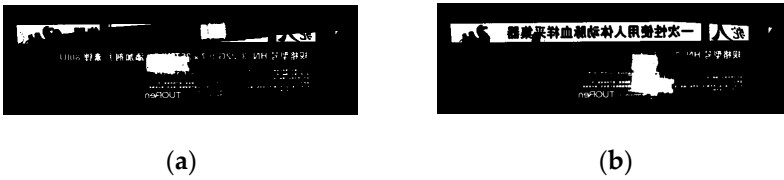

**Figure 6.** Threshold segmentation of the *S* component. (**a**) Not adhered; (**b**) adhered.

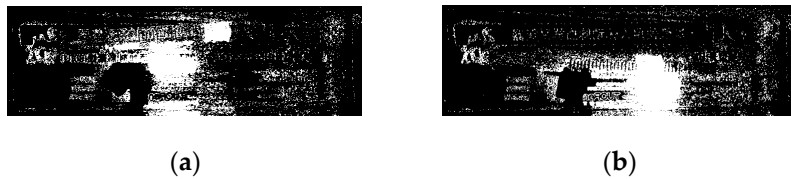

**Figure 7.** Segmentation of the *H* component. (**a**) Not adhered; (**b**) adhered.

By employing the cv2.setMouseCallback function in OpenCV, the maximum and minimum feature gray values in the V component were determined to be 149 and 72, respectively. Furthermore, a statistical analysis was conducted on the gray values of all pixels in the V component (illustrated in Figure 8). However, it was observed that the background gray values in the V component largely coincide with the feature gray values. Consequently, the V component cannot be effectively utilized for feature segmentation.

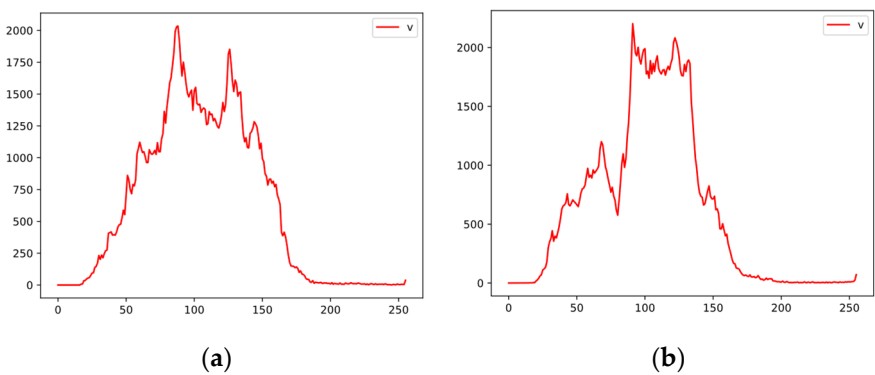

**Figure 8.** Histogram of gray values in *V* component. (**a**) Not adhered; (**b**) adhered.

Based on the aforementioned analysis results, when segmenting the features of the cone head seal and drawbar piston using HSV threshold segmentation, the upper threshold is set to np.array(109,198,255) and the lower threshold is np.array(96,125,0). The segmentation effect on the cone head seal and drawbar piston features is significant, as they can be separated from other background areas in the image. To reduce the interference around the features, morphological operations were performed on the image features. The processing result is shown in Figure 9.

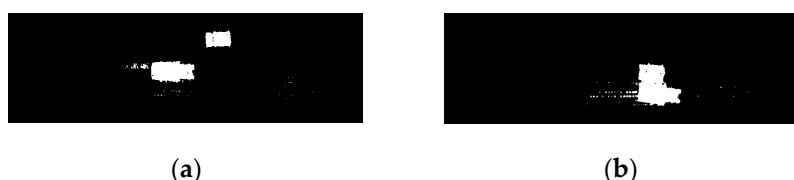

**Figure 9.** *Cont.*

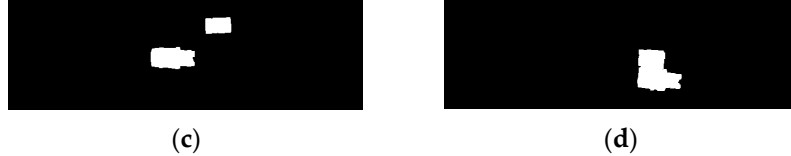

(c)                                                    (d)

**Figure 9.** Image processing result. (**a**) Non-adhesive comprehensive segmentation; (**b**) adhesive comprehensive segmentation; (**c**) non-adhesive morphological processing; (**d**) adhesive morphological processing.

Based on the morphology processing results above, upsampling is performed once, the area of connected regions is calculated using the cv2.contourArea function, and then the drawbar piston and cone head seal are classified based on the area of connected regions. When the area of connected regions in the image Area(m) is greater than a lower threshold $Area_{Tb1}$ and less than an upper threshold $Area_{Tb2}$, the drawbar piston exists, as indicated by expression (3). When the area of connected regions in the image Aare(m) is greater than a lower threshold $Area_{Tc1}$ and less than an upper threshold $Area_{Tc2}$, the cone head seal exists, as indicated by expression (4). When the area of connected regions in the image Area(m) is greater than an upper threshold $Area_{Tc2}$, it is determined that both the connecting drawbar piston and the cone head seal exist and are in an adhesion state, as indicated by expression (5). The lower threshold of the cone head seal area is greater than the upper threshold of the connecting drawbar piston area, as indicated by expression (6).

$$Area_{Tb1} < Area(m) < Area_{Tb2} \tag{3}$$

$$Area_{Tc1} \leq Area(m) \leq Area_{Tc2} \tag{4}$$

$$Area_{Tc2} \leq Area(m) \tag{5}$$

$$Area_{Tc1} > Area_{Tb2} \tag{6}$$

### 4.3. Identification of Blood Collection Needle Head

From Figure 3, it is evident that the blood collection needle head exhibits numerous other features with similar grayscale values in each component image. Despite attempts to adjust the upper and lower threshold values of the H, S, and V components in the cv2.inRange function module for HSV component segmentation, significant interference around the blood collection needle head feature persists. Consequently, accurately extracting the characteristics of the blood collection needle head becomes challenging, as depicted in Figure 10.

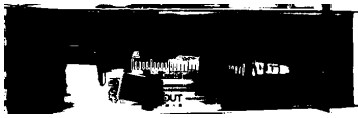

**Figure 10.** Segmentation result of blood collection needle head based on packaging ROI.

To address the above issue, a method based on locating the ROI of the blood collection needle head using key points of the drawbar piston is adopted. The main principle is to use the cv2.findContours function for contour detection of the segmented drawbar piston, and extract the top-left key point (x, y) of the contour. As the blood collection needle head is mounted at one end of the sample container and its shape, size, and vertical movement range are fixed, the key point of the drawbar piston can be cropped in the X-axis-positive direction. In order to ensure that each cropping area includes the features of blood collection needle head, the maximum projection $D_{max}$ of the distance between point (x, y) and the feature of blood collection needle head is selected as the cropping length in the direction of

X axis, that is, when the needle is parallel to X axis. In addition, to avoid cropping beyond the region of interest (ROI), the sum of the key point's horizontal coordinate x and $d_{max}$, denoted as $x_\alpha$, should be less than the ROI's horizontal size w, as shown in Equation (7). The ROI image has a non-active area for the blood collection needle in the vertical direction, so a length of d is cropped in both positive and negative directions of the Y axis, resulting in the Y axis coordinate points $y_{\alpha 1}$ and $y_{\alpha 2}$ used to crop the blood collection needle ROI, as shown in Equation (8).

$$x_{\alpha 2} = x + D_{max} < W \tag{7}$$

$$y_{\alpha 1} = d, y_{\alpha 2} = H - d \tag{8}$$

where the vertical height of the packaging ROI is h, and the coordinates for cropping the ROI are $(x, y_{\alpha 1})$ and $(x_{\alpha 2}, y_{\alpha 2})$ for the blood collection needle head ROI, as shown in Figure 11a.

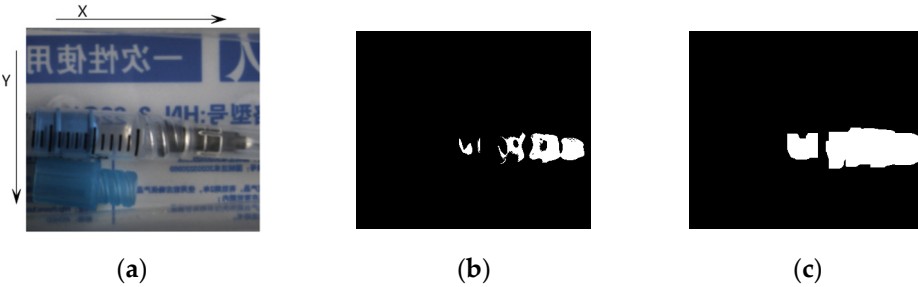

(**a**)　　　　　　　　(**b**)　　　　　　　　(**c**)

**Figure 11.** Blood collection needle head ROI segmentation process. (**a**) Blood collection needle head ROI; (**b**) threshold segmentation image; (**c**) morphological processing image.

After locating the ROI for the blood collection needle, it undergoes filtering with the cv2.Gaussianblur function to achieve a smoother image. Similar to other component segmentation methods, multiple trials were conducted to determine the optimal upper and lower threshold values. It was found that setting the upper threshold as np.array(40,40,78) and the lower threshold as np.array(0,0,30) yielded a more pronounced segmentation effect for the needle plug feature and other background regions in the image. Since the segmented features are relatively discrete, a dilation morphology operation is applied to connect them. The resulting segmentation is depicted in Figure 11b,c.

Due to the presence of irrelevant interference in the processed image, further determination of features based on the connectivity domain area and diagonal length is necessary. Firstly, the connected domain contours are detected using the cv2.findContours function. The top-left corner point $(x_i, y_i)$, horizontal projection $w_i$, and vertical projection hi of the connected domain contour are obtained to calculate the diagonal length D(i) of the feature using expression (9). By using the cv2.contourArea function to calculate the area of connected regions Area(i), if the diagonal length D(i) is greater than the lower threshold $d_{tg1}$ and less than the upper threshold $d_{tg2}$, and the area of the region, Area(i), is greater than the lower threshold $Area_{tf1}$ and less than the upper threshold $Area_{tf2}$, it is determined that a blood collection needle is present. Conversely, if the conditions are not met, it is determined as unqualified according to Equation (10).

$$D(i) = \sqrt{[x_i - (x_i + w_i)]^2 + [y_i - (y_i + h_i)]^2} \tag{9}$$

$$d_{Tg1} \leq D(i) \leq d_{Tg2}, Area_{Tf1} \leq Area(i) \leq Area_{Tf2} \tag{10}$$

## 5. Experiment

### 5.1. Experimental Equipment and Configuration

The experimental system consists of several components, including a loading mechanism for arterial blood collection needles, a loading mechanism for needle plugs, a loading mechanism for cone sealings, an arterial blood collector packaging machine, a PC equipped with an Intel Core i5-7300HQ quad-core processor, a color sensor, a PLC controller (Delta DVP28SV11S2), an Ethernet communication module (Delta DVPE-N_01), an industrial camera (Hikvision MV-CS050-10GC), an industrial lens (MV-CS050-10GC), a bar light source (MV-LLDS-192-38-W), a light source controller (MV-LEVD-60-4-SY), and a gigabit switch. The composition of the experimental system is shown in Figure 12.

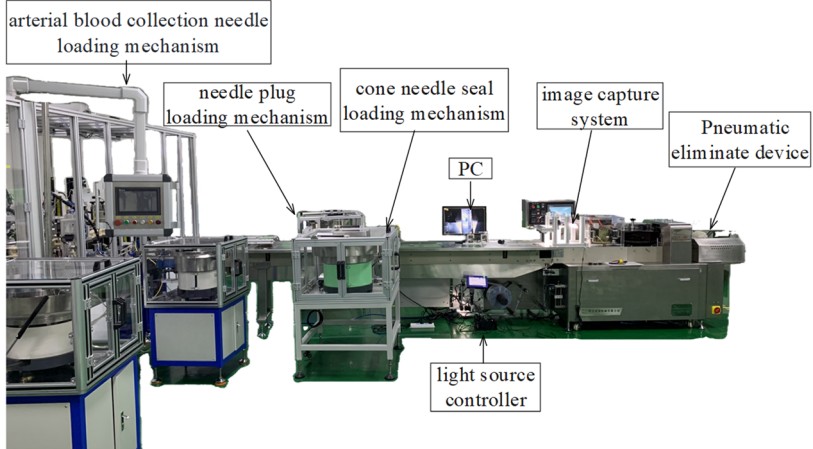

**Figure 12.** Test system composition.

The capture control system realizes the automation of image capture operations and provides PC-based image processing and result determination capabilities, as shown in Figure 13.

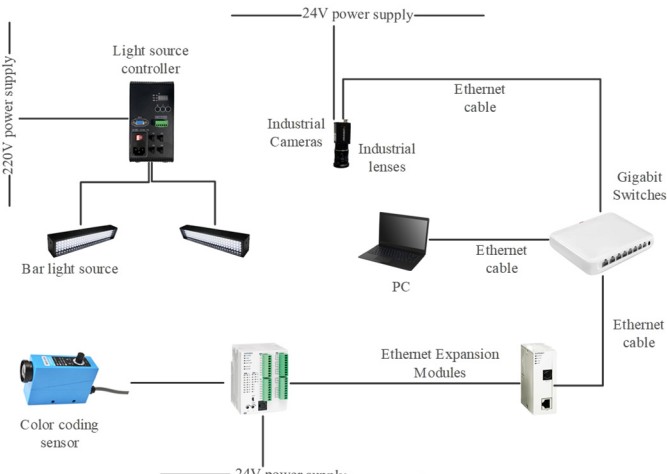

**Figure 13.** Capture control system structure.

In order to facilitate the adjustment of the camera and light source, we designed the mounting structure. The camera and light source are mounted on a 40 mm × 40 mm profile by anchor-type connectors so that their positions can be freely adjusted in X, Y and Z directions; they can also be rotated around the Y axis. The mounting structure is shown in Figure 14.

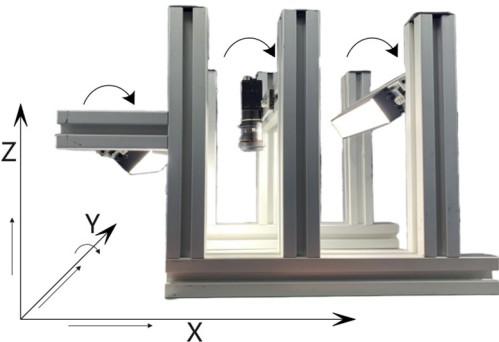

**Figure 14.** Camera and light source mounting architecture.

## 5.2. Experimental Methods and Related Parameters

When a package with a blue label passes through the color sensor detection area, the color sensor sends a detection signal to the PLC input terminal via I/O communication. After an internal delay relay of 85 ms in the PLC, the package to be detected arrives at the camera capture area and triggers the photo-taking relay. The PC reads the register status of the PLC relay through the Modbus-TCP communication protocol based on Ethernet. When the register status is ON, the PC captures one frame of real-time image for processing and judgment, specifically analyzing the image of the package to be detected. If the judgment result is deemed qualified, the product will pass smoothly. Otherwise, the PC writes the unqualified signal into the PLC, and the internal program of the PLC controls the pneumatic rejection device to remove the unqualified products.

Samples were collected based on the presence of materials, with 200 samples in each category. The connected area or diagonal length value of the connected domain was measured in the sample image to determine the threshold limit. Based on the statistical results of each feature, a line chart was drawn (as shown in Figure 15), with the maximum connected domain area of the needle plug being 3676 and the minimum being 2310. The maximum connected domain area of the cone seal is 18,694 and the minimum is 15,491. The maximum value of the drawbar piston is 10,828, and the minimum is 8978. When the cone seal adheres to the drawbar piston, the maximum value is 30,039 and the minimum is 25,877. A diagonal length judgment was added for the blood collection needle head, with a maximum area of 10,909 and a minimum of 6397. The maximum diagonal length is 214.41082 and the minimum is 130.60245. The maximum and minimum values of each feature's connected domain are shown in Table 1.

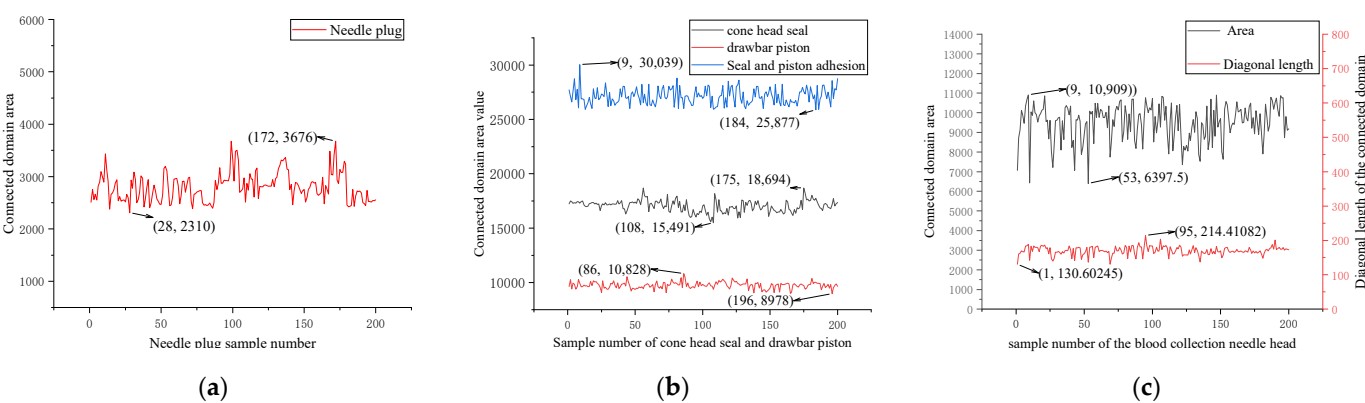

**Figure 15.** Line chart of connected domain area or diagonal length statistics. (**a**) Needle plug; (**b**) drawbar piston and the cone seal; (**c**) blood sampling needle.

**Table 1.** Maximum value of each connected domain feature.

| Connected Domain Determination Type | Blood Collection Needle Head | Needle Plug | Cone Head Seal | Drawbar Piston | Cone Head Seal and Drawbar Piston Adhesion |
|---|---|---|---|---|---|
| Maximum area value | 10,909 | 3676 | 18,694 | 10,828 | 30,039 |
| Minimum area value | 6397 | 2310 | 15,491 | 8978 | 25,877 |
| diagonal length maximum value | 214.41082 | | | | |
| diagonal length minimum value | 130.60245 | | | | |

The area or diagonal length of the connected domain in the samples was statistically analyzed. To ensure that all features are detected as much as possible, the threshold range should include the maximum and minimum values obtained from the statistics.

*5.3. Experimental Results and Analysis*

Through testing the samples, the threshold values were determined as follows: for the needle plug, $Area_{Ta} = 2079$; for the drawbar piston, $Area_{Tb1} = 8080$ and $Area_{Tb2} = 11,911$; for the cone head seal, $Area_{Tc1} = 13,942$ and $Area_{Tc2} = 20,563$; for the blood collection needle head, $Area_{Tf1} = 5758$, $Area_{Tf2} = 12,000$, $d_{Tg1} = 104$, and $d_{Tg2} = 257$, which produced good results.

First, based on a set threshold for specific characteristics, the missing component types are divided into five categories. A total of 200 experimental samples are tested for each category to conduct inspection tests on arterial blood sample collector components. The average processing time and false detection rate of image processing on the PC are recorded during the testing process. The system detection results are shown in Table 2.

**Table 2.** Detection results.

| Needle Plug | Drawbar Piston | Cone Head Seal | Blood Collection Needle | Number of Samples | Detection Result | | Noise Factor | Image Processing Average Time Consumption |
|---|---|---|---|---|---|---|---|---|
| | | | | | Qualified | Unqualified | | |
| presence | presence | presence | presence | 200 | 198 | 2 | 0.1% | 0.039 |
| absence | presence | presence | presence | 200 | 0 | 200 | 0 | 0.038 |
| presence | absence | presence | presence | 200 | 0 | 200 | 0 | 0.034 |
| presence | presence | absence | presence | 200 | 0 | 200 | 0 | 0.039 |
| presence | presence | presence | absence | 200 | 0 | 200 | 0 | 0.038 |

Secondly, the effectiveness of the image processing algorithm in practice is then verified by five types of samples. The number of samples packed in each category is 100, and the products are divided into qualified and unqualified products according to the absence of components at the time of packing. The packaging validation results are shown in Table 3.

In actual production, the manual recognition of 1000 sample packages takes an average of 1.5 s per item, with an accuracy rate of 97.5%. Efficiency and accuracy will both decline as fatigue increases. Based on machine vision for identifying packaging defects, according to the detection results in Table 2, the detection accuracy for qualified products can reach 99%. The main reason for false detections is the non-flat posture of the components placed by the feeding mechanism, which causes changes in the detected feature shapes. By optimizing the feeding mechanism, the accuracy can be further improved. For defective products, the false detection rate is 0%. The processing time for each image can be controlled within 40 ms, while the production time for each package is approximately 1000 ms. In addition, according to the actual production detection results in Table 3, it can be seen that

the recognition rate of qualified products is high, and all of them can effectively detect the unqualified products. This indicates that the image recognition algorithm has good application performance. Therefore, it fully meets the detection requirements for products.

**Table 3.** Product Validation Results.

| Number of Qualified Products | Number of Non-Conforming Products | | | | Sample Size | Detection Result | | Recognition Rate |
|---|---|---|---|---|---|---|---|---|
| | Missing Needle Plug | Missing Drawbar Piston | Missing Cone Head Seal | Missing Blood Collection Needle | | Qualified | Unqualified | |
| 80 | 5 | 5 | 5 | 5 | 100 | 80 | 20 | 100% |
| 75 | 10 | 5 | 5 | 5 | 100 | 75 | 25 | 100% |
| 70 | 10 | 10 | 5 | 5 | 100 | 69 | 31 | 99% |
| 65 | 10 | 10 | 10 | 5 | 100 | 65 | 35 | 100% |
| 60 | 10 | 10 | 10 | 10 | 100 | 59 | 41 | 99% |

## 6. Conclusions

Based on OpenCV, using gray value correlation functions as indicators for arterial blood sample collector packaging images, template matching can effectively locate the ROI of the packaging image. In the HSV color space, by selecting appropriate upper and lower threshold values, threshold segmentation can be applied to the features within the packaging ROI. After filtering and morphological processing, it is possible to accurately calculate the area and diagonal length of the connected domain. In the HSV color space, the appropriate upper and lower thresholds are selected to segment the features in the packaged ROI. After filtering and morphological processing, the connected domain area and diagonal length can be accurately calculated. Based on the connected domain area criterion, the needle plug, sample storage container, and cone head seal can be effectively identified. By using both the connected domain area and diagonal length criteria, the blood collection needle head can be effectively identified.

In the future, by expanding the threshold and other related parameters, supplementing judgment criteria, and integrating with systems, the image processing algorithm studied in this paper can be adapted to recognize the packaging components of the arterial blood sample collectors of more models. Meanwhile, the recognition process can be extended to other industrial scenarios based on color recognition, but it is necessary to add or reduce the relevant image processing algorithms and debug the threshold range to achieve a stable result according to the actual situation.

**Author Contributions:** Conceptualization, Z.S.; methodology, Z.S.; software, Q.W.; validation, Q.W.; formal analysis, Z.S.; investigation, Q.W.; resources, Z.S.; data curation, Z.L.; writing—original draft preparation, Q.W. and Z.L.; writing—review and editing, Z.S.; visualization, Z.L.; supervision, Z.S.; project administration, Z.S.; funding acquisition, Z.S. All authors have read and agreed to the published version of the manuscript.

**Funding:** This research received no external funding.

**Institutional Review Board Statement:** Not applicable.

**Informed Consent Statement:** Informed consent was obtained from all subjects involved in the study.

**Data Availability Statement:** Not applicable.

**Acknowledgments::** We would like to express our deepest gratitude to all those who have supported and contributed to this work, making it possible to achieve its completion. We are especially thankful to the Intelligent Manufacturing Equipment Department at Blue Light Lab, Tuoren Medical Company Limited, Xinxiang, China, for providing the necessary resources and facilities for this project. Additionally, we extend our heartfelt appreciation to Han Zhengfeng from the company, whose expertise and guidance significantly improved the quality of this project. Furthermore, we are

grateful to our team members for their valuable insights and feedback during the development of this work. Their constructive criticism and thoughtful suggestions played a crucial role in shaping the final outcome. This work would not have been possible without the collective contributions and encouragement from all these individuals and entities. We are truly grateful for their support.

**Conflicts of Interest:** The authors declare no conflict of interest.

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
