# Peer review of "Machine Vision Algorithm for Identifying Packaging Components of HN-3 Arterial Blood Sample Collector"

_applsci, doi:10.3390/app13148450_

Round 1

Reviewer 1 Report

The Authors has reported an excellent work on machine vision based algorithm for detection of packaging components in the HN-3 arterial blood sample collector. The image processing algorithm for identification of the components inside the product packaging was effectively utilized. The effectiveness of the developed image processing technique was experimentally evaluated. A systematic experimental approach was adapted to analyse the accuracy of the developed image processing methodology. The works carried out by the authors have significant relevance in the medical device production industry particularly in the packing sections. Considering the scope and quality of the manuscript the reviewer is of the opinion that the manuscript can be accepted got publication in applied sciences in the current form.

Author Response

Many thanks to the reviewers for their approval of this article.

Reviewer 2 Report

The article addressess missing components in the packaging bag of arterial blood sample. The contents are well organized. However, there are a few concerns to be addressed by the authors. 

It would be more clear if the autors provide the size of the problem. How often the addressed issue happens per hundred samples? 

What is the novelty? Highlight your contributions before starting the methodology (at the end of section 1). 

many mathematical relations given in the paper (Foe example equation 1 to 5) well known equations. Such equations need not be included. If the authors have changed any value or expression, such equations can be included with appropriate explanation. 

How do the authors match the pixel locations of template image and input image? This needs a clear expalnation. 

Normally, thresholding is affected by the image quality. How did the authors handle intensity and color variations? What is the image resolution? Was the image captured in controlled environment? If yes, did the authors check the robustness of the algorithm by varying the illumination and brightness?

Provide sufficient details about image capturing system, since the entire methodology is based on image processing. 

Explain about the noise factor.

Mention any future scope if any.

Author Response

Thank you very much for the reviewer's suggestions. I have carefully read them and provided a response to each question.Please review by the reviewers.Please see the attachment.

Question 1:It would be more clear if the autors provide the size of the problem. How often the addressed issue happens per hundred samples? 

The following content is the answer to the question and has been added to the manuscript of the paper.

Verify the effectiveness of image processing algorithms in practice using five types of samples. Each type of sample consists of 100 packages, and the products are classified as either qualified or unqualified based on the presence of ingredient deficiencies during packaging. The packaging verification results are shown in the table below.

Number of qualified products

Number of non-conforming products

Sample size

Detection result

Recognition rate

Missing needle plug

Missingdrawbar piston

Missing Cone head seal

Missing Blood collection needle

qualified

unqualified

80

5

5

5

5

100

80

20

100%

75

10

5

5

5

100

75

25

100%

70

10

10

5

5

100

69

31

99%

65

10

10

10

5

100

65

35

100%

60

10

10

10

10

100

59

41

99%

From the actual production inspection results of this table, we can see that the recognition rate of qualified products is high, and all of them can effectively detect the unqualified products. It shows that the image recognition algorithm has good application performance. Therefore, it fully meets the requirements of product detection.

Question 2:What is the novelty? Highlight your contributions before starting the methodology (at the end of section 1).

The following content is the answer to the question and has been added to the manuscript of the paper.

This algorithm utilizes advanced image processing techniques to effectively analyze and process images, resulting in precise identification of components. The evaluation of experimental results has showcased the algorithm's exceptional accuracy and rapid processing time, thereby establishing its feasibility and practicality for real-world applications. The outcomes of this research hold considerable significance for the arterial blood gas analysis market in China, offering medical device manufacturers and healthcare institutions a valuable opportunity to enhance product quality and work efficiency. By improving the accuracy and efficiency of packaging component recognition, this algorithm has laid the foundation for substantial progress in the industry.

Question 3:many mathematical relations given in the paper (Foe example equation 1 to 5) well known equations. Such equations need not be included. If the authors have changed any value or expression, such equations can be included with appropriate explanation. 

Have removed the formulas suggested by the reviewer and made modifications to the textual content related to the formulas in the manuscript.

Question 4:How do the authors match the pixel locations of template image and input image? This needs a clear expalnation. 

An explanation has been provided for this issue, and the relevant content has been added to the manuscript. In order to maintain coherence in the context, slight modifications have been made to the surrounding content.The following is the answer to this question:

In template matching, pixel-level matching is used to determine the position of a template image within a target image. By sliding the template image over different positions of the target image and calculating the similarity between them, the best-matching pixel position can be found.

Specifically, the template and target images are first converted to grayscale for pixel-level matching. The template image is then slid over every possible position in the target image, and the similarity between pixels at each sliding position is measured using methods like normalized squared difference. This involves calculating the squared differences and normalizing them to obtain similarity values.These similarity values are organized into a matching result matrix, which has the same size as the target image. Each element of the matrix represents the similarity measure at the corresponding position. The best matching position in the result matrix is typically determined by finding the minimum or maximum value, which corresponds to the vertex or center position of the template image in the target image.Pixel-level matching allows for accurate localization of the template image within the target image, enabling ROI extraction.

Question 5:Normally, thresholding is affected by the image quality. How did the authors handle intensity and color variations? What is the image resolution? Was the image captured in controlled environment? If yes, did the authors check the robustness of the algorithm by varying the illumination and brightness?

The initial captured image resolution is 2448×2048, based on which feature extraction is performed. The threshold range usually needs to be readjusted for luminance and color changes in the image. During the debugging process, it was found that this threshold range can still fully extract features with less interference even for slight changes in brightness and color in the image. In addition, the environment in which the images are captured is controlled, and the mounting position and light intensity remain largely stable. During the debugging process, it is found that the robustness of the algorithm is not affected even when slight illumination and luminance changes occur.

Question 6:Provide sufficient details about image capturing system, since the entire methodology is based on image processing. 

Based on the reviewers' suggestions, I have made relevant additions to the capture system in the paper.

Question 7:Explain about the noise factor.

The following is my explanation of image noise, which has been added to the manuscript of the paper.

Image noise refers to unwanted interference or random disturbances present in digital images. It can be introduced during the processes of image acquisition, transmission, storage, and processing, resulting in a degradation of image quality and loss of information. Therefore, prior to performing feature segmentation on the ROI image, it is usually necessary to apply image filtering techniques to reduce the interference caused by noise points and improve the accuracy of subsequent image segmentation.

Question 8:Mention any future scope if any.

The following content is a vision of the future and has been added to the manuscript of the paper.

In the future, by expanding threshold and other related parameters, supplementing judgment criteria, and integrating with systems, the image processing algorithm studied in this paper can be adapted to recognize packaging components of arterial blood sample collectors of more models.Meanwhile, the recognition process can be extended to other industrial scenarios based on color recognition, but it is necessary to add or reduce the relevant image processing algorithms and debug the threshold range to achieve a stable result according to the actual situation.

Reviewer 3 Report

 Machine Vision Algorithm for Identifying Packaging Components of HN-3 Arterial Blood Sample Collector” by Shang et al. has developed a PyCharm integrated environment for identifying the components inside the HN-3 arterial blood sample collector’s package. The experimental results have indicated good accuracy with a fast processing time of the implemented methods. This study is interesting from the engineering aspect, the current results are presented in a reasonable way, and the complete work does have an impact on the market of arterial blood gas analysis in China. Here are my several comments to further improve the manuscript.

1. The authors have discussed a specific method for the HN-3 arterial blood sample collector’s package. Will the identification pipeline be possible to be extrapolated to other industrial scenarios? Please elaborate.

2. How is the robustness of the method given more complex imaging environments? Will the illumination, orientation, etc. affect the accuracies and efficiencies of such a method? If so, how can the results be improved? Please discuss.

Please further polish the English writing.

Author Response

Thank you very much for the reviewer's suggestions. I have carefully read them and provided a response to each question.Please review by the reviewers.Please see the attachment.

Question 1:The authors have discussed a specific method for the HN-3 arterial blood sample collector’s package. Will the identification pipeline be possible to be extrapolated to other industrial scenarios? Please elaborate.

The following content is the answer to that question, part of which has been added to the conclusion of the manuscript.

The recognition process is based on image processing and analysis techniques, designed and optimized specifically for the packaging components of the HN-3 arterial blood sample collector. While this method has shown good performance in handling the HN-3 collector packaging components, its extension to other industrial scenarios needs to consider the following factors:

1.Discriminability of features: The effectiveness of the recognition process relies on the discriminability of the features. When extending the method to other industrial scenarios, it is important to ensure that the objects or components to be recognized possess sufficient feature information for accurate image processing and analysis. If the features are not distinct enough or prone to interference, additional techniques or methods may need to be employed to enhance their discriminability. The recognition process can be easily extended to other industrial scenarios based on color recognition, but adjustments to the image processing algorithms and threshold ranges may be required based on the specific circumstances to achieve a stable performance.

2.Image acquisition devices and environment: The success of the recognition process also depends on the quality of the image acquisition devices and the environment. Different industrial scenarios may have varying environmental conditions and acquisition devices, such as lighting conditions and camera resolutions. When promoting the recognition process, appropriate adjustments and optimizations should be made based on the specific scenario to ensure that the image quality meets the requirements and provides accurate feature information.

3.Customization for specific problems: The recognition process may require customized adjustments based on different industrial scenarios. Various industrial scenarios may involve different object shapes, sizes, textures, and other features. Therefore, when extending the recognition process, customized algorithm and method designs may be necessary to address specific requirements of the particular scenario.

In summary, the extension of the recognition process to other industrial scenarios is possible but requires a detailed analysis and customized adjustments according to the specific demands of the scenario. By selecting suitable algorithms and technologies and considering the practical image acquisition devices and environmental conditions, the recognition process can be successfully applied in other industrial scenarios to achieve automated recognition and detection of different objects or components.

Question 2:How is the robustness of the method given more complex imaging environments? Will the illumination, orientation, etc. affect the accuracies and efficiencies of such a method? If so, how can the results be improved? Please discuss.

While the algorithm exhibits high recognition rates for the images in the manuscript, its robustness may be affected by factors such as lighting and orientation in more complex image environments. These factors can lead to more significant variations in brightness and color in the images, requiring further adjustment of threshold ranges to adapt to these changes.Furthermore, the following improvements can be made to enhance the accuracy and efficiency of the algorithm in complex environments:

1.Dynamic threshold range adjustment: Adaptively adjust the threshold range based on the degree of brightness and color variations in the image to better extract features. Techniques such as the Otsu algorithm or adaptive Gaussian thresholding can be employed.

2.Introduce additional features: In addition to HSV threshold segmentation, consider incorporating other feature extraction methods to increase the robustness of the algorithm. These additional features can be further analyzed and processed after HSV threshold segmentation.

3.Utilize machine learning methods: Explore the use of machine learning algorithms to train models for recognizing and differentiating features in different environments. Deep learning methods such as convolutional neural networks (CNNs) can be employed to train on a large number of samples, thereby improving the robustness and accuracy of the algorithm.

When improving the algorithm, thorough experimentation and validation should be conducted to ensure that the enhanced approach performs effectively in diverse environments.
